# Both pathogen and host dynamically adapt pH responses along the intestinal tract during enteric bacterial infection

**Sarah E. Woodward[1,2], Laurel M. P. Neufeld[1], Jorge Peña-Díaz[1,2], Wenny Feng[2], Antonio Serapio-Palacios[1,2], Isabel Tarrant[1,2], Wanyin Deng[2], B. Brett Finlay**[1,2,3]*

**1** Department of Microbiology and Immunology, University of British Columbia, Vancouver, Canada, **2** Michael Smith Laboratories, University of British Columbia, Vancouver, Canada, **3** Department of Biochemistry and Molecular Biology, University of British Columbia, Vancouver, Canada

* bfinlay@interchange.ubc.ca

**Data Availability Statement:** All relevant data are within the paper and its Supporting Information

## Abstract

Enteric pathogens navigate distinct regional microenvironments within the intestine that cue important adaptive behaviors. We investigated the response of *Citrobacter rodentium*, a model of human pathogenic *Escherichia coli* infection in mice, to regional gastrointestinal pH. We found that small intestinal pH (4.4–4.8) triggered virulence gene expression and altered cell morphology, supporting initial intestinal attachment, while higher pH, representative of *C. rodentium*'s replicative niches further along the murine intestine, supported pathogen growth. Gastric pH, a key barrier to intestinal colonization, caused significant accumulation of intra-bacterial reactive oxygen species (ROS), inhibiting growth of *C. rodentium* and related human pathogens. Within-host adaptation increased gastric acid survival, which may be due to a robust acid tolerance response (ATR) induced at colonic pH. However, the intestinal environment changes throughout the course of infection. We found that murine gastric pH decreases postinfection, corresponding to increased serum gastrin levels and altered host expression of acid secretion-related genes. Similar responses following *Salmonella* infection may indicate a protective host response to limit further pathogen ingestion. Together, we highlight interlinked bacterial and host adaptive pH responses as an important component of host–pathogen coevolution.

## Introduction

Gastrointestinal pH is a key component of gut physiology, known to vary across regions of the intestine [1,2]. In humans, intestinal pH is characterized by a progressive increase from the duodenum to the distal ileum (pH 6.0 to 7.5), a drop at the cecum to pH 6.4, followed by increasing pH from the proximal to the distal colon (pH 7 to 7.5; Fig 1A) [1–3]. In mice, reported intestinal pH is much lower, with the intestinal contents of female BALB/c mice ranging from 2.98 in the stomach to 5.24 in the ileum (Fig 1A) [4]. In both species, maintaining intestinal pH is important to homeostasis and gut function. Disruption to intestinal ion transport is associated with spontaneous onset of colitis symptoms in mice and increased

files. Source data is specifically found in supporting file S1 Data which accompanies this manuscript.

**Funding:** This work was supported by grants from the Canadian Institutes of Health Research (CIHR; cihr-irsc.gc.ca) to BBF (FDN-159935). SEW is a CIHR CGS-D Graduate Scholar (GSD-154171) and was supported by a University of British Columbia (UBC; ubc.ca) Four Year Fellowship and Dmitry Apel Memorial Scholarship. The funders had no role in study design, data collection and analysis, decision to publish, or preparation of the manuscript.

**Competing interests:** The authors have declared that no competing interests exist.

**Abbreviations:** ATR, acid tolerance response; CBA, cytometric bead array; CFU, colony-forming unit; DMEM, Dulbecco's Modified Eagle Medium; DSS, dextran sulfate sodium; FD, functional dyspepsia; GI, gastrointestinal; HA, host-adapted; LB, lysogeny broth; MOI, multiplicity of infection; OD, optical density; PBS, phosphate buffered saline; ROS, reactive oxygen species; T3SS, type III secretion system.

susceptibility to induced colitis [5–7]. pH also influences which microbial species colonize the gut, with mildly acidic pH resembling proximal colon conditions inhibiting the growth of *Bacteroides* spp. [8–10]. Therefore, maintenance of gut pH is important to preventing overgrowth of colonizing bacteria, whether commensal or pathogenic [8,9].

Intestinal pH is often overlooked with regard to enteric infection, despite its potential role in affecting infection dynamics, host–pathogen interactions, and possibly even therapeutic success. By definition, diarrheal disease agents interact with the gastrointestinal (GI) environment to establish infection. Pathogenic *Escherichia coli*, including enteropathogenic *E. coli* (EPEC) and enterohemorrhagic *E. coli* (EHEC), are a diverse group of diarrheal disease agents which result in diarrhea, fever, vomiting, and even death in infected hosts [11,12]. *Citrobacter rodentium*, a related pathogen that shares similar virulence factors, is used to model murine EPEC/EHEC infection [13–15]. *C. rodentium* naturally infects mice, causing transmissible murine colonic hyperplasia (a form of colitis) with diarrheal disease symptoms [16]. All 3 pathogens are transmitted through the fecal-oral route and use a type III secretion system (T3SS) for successful colonization [13,17–21], translocating protein effectors directly into host cells for bacterial control over host cell processes [22].

It is clear that EPEC/EHEC/*C. rodentium* respond to intestinal pH during host colonization. In particular, the extreme acidic conditions of the gastric environment are a major bottleneck to both *C. rodentium* and EHEC colonization [23–26]. Furthermore, within the small intestine *C. rodentium* responds to alkaline bicarbonate ions secreted by the host to neutralize acidic chyme by up-regulating genes involved in initial epithelial attachment [27,28]. Despite this work, much remains unclear about how EPEC/EHEC/*C. rodentium* respond to regional differences in gastrointestinal pH. Furthermore, while the GI environment is known to change as a consequence of diarrheal disease, little investigation has been done to profile intestinal pH postinfection [29].

This study investigates the behavior and virulence of *C. rodentium* in response to variations in host intestinal pH. We profiled pH across the intestine of one of the most commonly used laboratory mouse strains (C57BL/6) to define a range of physiologically relevant pH encountered by enteric pathogens upon initial host contact. We identified pH-specific regulation of the T3SS and epithelial attachment by *C. rodentium*, as well as significant changes to stress

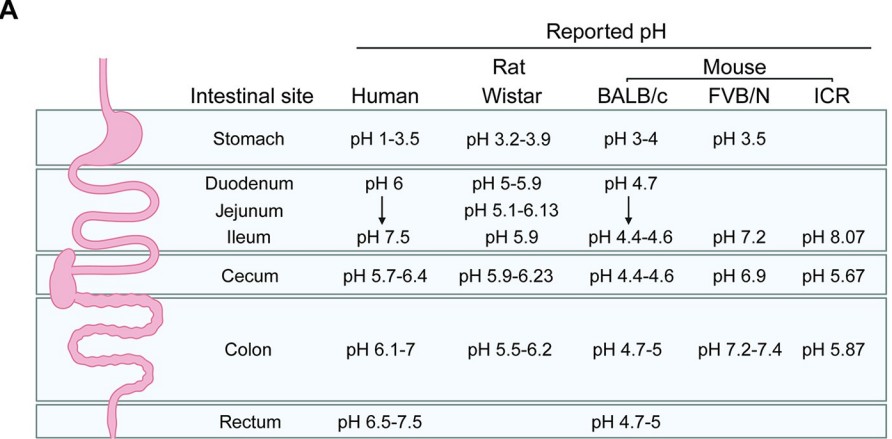
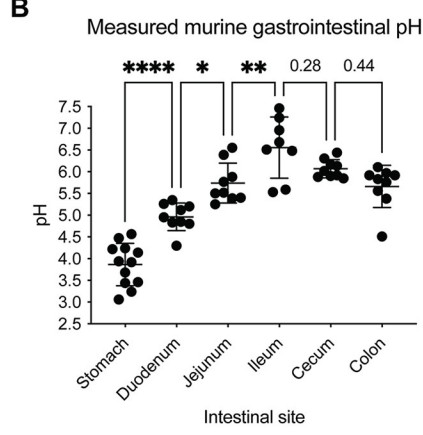

**Fig 1. Gastrointestinal pH varies across gut regions.** (A) Average lumenal pH of gut regions in murine and human gastrointestinal tracts, as reported in the literature [1–4,44–46]. (B) Lumenal pH measured across the gut of uninfected C57BL/6 mice (N = 8–12). Error represents mean +/− SD. Statistical analysis represents a one-way ANOVA with Tukey's multiple comparisons test. Summary data displayed in Fig 1 can be found in S1 Data. Created with BioRender.com.

adaptation. Additionally, within-host adaptation increased *C. rodentium*'s acid tolerance, stimulated by colonic pH conditions. The pH of the gut also changed during established *C. rodentium* infection, with a significant drop in gastric pH which correlated with pathogen burden. This effect was consistent across additional models of intestinal damage, suggesting a possible host defense mechanism against further pathogen ingestion. Together, these data emphasize the interplay between pathogens and host intestinal pH in the context of enteric infection.

## Materials and methods

### Ethics statement

Animal experiments were performed in accordance with the guidelines of the Canadian Council on Animal Care and the University of British Columbia (UBC) Animal Care Committee. Approval for the work carried out in this study was granted by the UBC Animal Care Committee according to Animal Care Protocols A20-0187 and A17-0228.

### Bacterial strains and culture conditions

Bacterial strains are listed in S1 Table. Unless otherwise noted, *C. rodentium* strain DBS100 was used in all assays [30]. A fresh bacterial stock was used for all assays as follows: 2 days in advance bacteria were streaked onto a fresh LB agar plate from frozen (−70˚C in glycerol) and grown overnight at 37˚C. A single colony was re-inoculated into LB (lysogeny broth, Sigma) broth and grown overnight at 37˚C, shaking. To test the response of bacteria to gastrointestinal pH, LB broth or DMEM (Dulbecco's Modified Eagle Medium, Hyclone) was adjusted to gastrointestinal pH by the addition of 37% HCl or 40% NaOH. Due to the high buffering capabilities of DMEM, media were pH-adjusted fresh for use within 12 h. pHs tested were: 3.5, 4.0, 4.4, 4.8, 5.7, 6.0, 6.4, 7.0, and 7.5. To test the effect of pH on human pathogenic *E. coli*, EHEC strain EDL933 [31] and EPEC strain E2348/69 [32] were used, as well as *Salmonella enterica* serovar Typhimurium strain CDC 6516–60 (ATCC 14028).

### *ldcC* mutant construction

The *sacB* gene-based allelic exchange method and the suicide vector pRE112 [33] were used to generate a chromosomal in-frame deletion mutant of *ldcC* in *C. rodentium* strain DBS100. The immediate upstream and downstream flanking regions of *ldcC* were amplified by PCR using primers listed in S2 Table. Purified DNA fragments were ligated into pRE112 by Gibson assembly. The suicide vector was transformed into *E. coli* strain MFD*pir* [34] by electroporation, which was used as the donor strain to transfer the plasmid by conjugation into *C. rodentium*. After sucrose selection, colonies resistant to sucrose and sensitive to chloramphenicol were screened, and then further verified, for *ldcC* deletion by PCR using primers Check_ldcC_F and Check_ldcC_R.

### *In vivo* pH measurements and mouse infections

As mentioned above, animal experiments were performed in accordance with the guidelines of the Canadian Council on Animal Care and the University of British Columbia (UBC) Animal Care Committee according to Animal Care Protocols A20-0187 and A17-0228. C57BL/6 and BALB/c mice were ordered from Jackson Laboratory (Bar Harbor, Maine, United States of America) and maintained in a specific pathogen-free facility at UBC on a 12-h light-dark cycle. Mice were allowed to acclimatize to the facility for 1 week following arrival. All pH measurements and infection models were run after the acclimatization period.

pH measurements were performed on fresh, undiluted intestinal content, using an Elite pH Spear Pocket Tester (Thermo Scientific). Measured pH represents the average of 2 to 3 readings per sample. All bacterial colonization experiments were done in 7-week-old female C57BL/6 mice fed the Picolab Rodent Diet 20 chow (Cat. no. 5053; LabDiet). Mice were monitored daily throughout the 4 to 8 day infections for weight loss and clinical symptoms.

*C. rodentium* infections were carried out by gavaging mice orally with $10^8$ colony-forming units (CFUs) of *C. rodentium* DBS100 from overnight culture. Mice were euthanized at experimental endpoint by isoflurane anesthesia followed by carbon dioxide inhalation. Bacterial burden was determined by collecting colon samples into 1 mL of reduced phosphate-buffered saline (PBS) and homogenized in a FastPrep-24 (MP Biomedicals) at 5.5 m/s for 2 min. Sample homogenate was diluted for plating on MacConkey agar (Difco) and incubated for 18 to 20 h at 37°C before counting bacterial colonies.

Chemically induced colitis was triggered by exposure to 3% dextran sulfate sodium (DSS) in the drinking water for 4 days before being placed on regular drinking water. Mice were euthanized at 5 days posttreatment onset.

Colonization with both avirulent *C. rodentium* Δ*escN* [35] and commensal *E. coli* Mt1B1 [36] were carried out by oral gavage with 2.5–4 × $10^8$ CFU from overnight culture, as determined by retrospective plating. Control WT-infected mice were inoculated at the same dosage. *S.* Typhimurium (ATCC 14028) infections were carried out by oral gavage with 1 × $10^6$ CFU of *S.* Typhimurium from overnight culture. Fecal samples were processed for CFU enumeration as described above.

## Analysis of rodent diets

To compare standard rodent diets, mice were fed either the Teklad Global 18% Protein Rodent Diet (Cat. no. 2918; Envigo) or Picolab Rodent Diet 20 (Cat. no. 5053; LabDiet), over a period of 2 weeks before measurement. Mouse chow pH was measured by dissolving pellets in 3 mL deionized water per 100 mg.

## Bacterial growth at gastrointestinal pH

Bacterial growth at gastrointestinal pH was determined by inoculating bacteria at an optical density at 600 nm ($OD_{600}$) of 0.005 from an overnight culture into 250 μL total volume of the media of interest (LB or DMEM base media adjusted to gastrointestinal pH). Cultures were incubated at 37°C with agitation for 20 h in a Synergy H1 plate reader (Biotek) with $OD_{600}$ measurements at 10-min intervals.

## Cell culture assays

Murine rectal carcinoma cell line CMT-93 (ATCC CCL-223) and human adenocarcinoma cell line Caco-2/TC7 were maintained in DMEM supplemented with 10% fetal bovine serum, 1% Glutamax, and 1% nonessential amino acids. Cells were used between passages 4 to 10. Cells were seeded in 12-well plates at 90,000 cells/well and incubated for 48 h prior to infection. Confluent CMT-93 cells were washed once with PBS before infection. The *C. rodentium* inoculum was prepared by subculturing 1:40 from overnight culture to mid-log phase (3.5 h, 37°C, shaking) in either LB or DMEM media adjusted to intestinal pH. Subcultures were washed and resuspended in cell culture media at multiplicity of infection (MOI) 100. Infected plates were centrifuged (1,000 rpm, 5 min) to synchronize the infection before incubation for 4 h at 37°C and 5% $CO_2$. Infected plates were washed 5 times with PBS before detachment using 0.1% Triton X-100 in PBS for dilution and plating on neutral LB agar. Adherent bacterial CFU

were quantified and data were normalized to exact inoculum dosage (determined by retrospective plating) to account for differences between pH conditions.

Evaluation of actin pedestal formation was performed as follows. Caco-2/TC7 and CMT-93 cells were cultured in a 6-channel slide (Cat. No. 80606; ibidi) to 80% confluence before infection as before with *C. rodentium* preconditioned to mid-log phase at pH 4.8 or pH 7.0. Following the 4-h infection, cells were washed with PBS, fixed with 4% paraformaldehyde for 20 min, permeabilized with 0.1% of Triton X-100 for 10 min, and blocked with 3% BSA–PBS for 30 min. Cells were immunostained using a polyclonal rat anti-Tir antibody [17] overnight, followed by a goat anti-rat IgG (H+L) secondary antibody conjugated to Alexa Fluor 568 (Cat. no. A-11077; Thermo Fisher Scientific) for 45 min. F-actin was stained with Alexa Fluor 488 phalloidin (Cat. no. A12379). Nuclei and bacteria were stained with DAPI. Cells were imaged with a Leica Stellaris 5 confocal microscope using a 63× objective and analyzed using LAS-X software.

## Bacterial and eukaryotic gene expression analysis

*C. rodentium* was subcultured for 3.5 h 1:40 in LB base media adjusted to gastrointestinal pH, as before. After subculture samples were preserved in RNAprotect Bacterial Reagent (Qiagen) and stored at −70˚C until extraction. Bacterial RNA extractions were done using the RNAse-Free DNase Set (Cat. no. 79254, Qiagen) as per the manufacturer's instructions. A QuantiTect Reverse Transcription Kit (Cat. no. 205313, Qiagen) was used for cDNA synthesis and Quantitative RT-qPCR was performed using a QuantiNova SYBR Green RT-PCR Kit (Cat. no. 208056) in combination with primers (S3 Table) for target gene mRNA. *dnaQ* was used as an endogenous control. Data were normalized to efficiency of primers by target gene (91.6% to 103.5% for all primers used).

Eukaryotic RNA was isolated from mouse tissue samples (stomach and colon) stored in RNAprotect Tissue Reagent (Qiagen) at −70˚C before extraction using a GeneJET RNA Purification Kit (Cat. no. K0731, Thermo Scientific). cDNA synthesis and Quantitative RT-qPCR was performed as described above using various primer pairs (S3 Table). β-2-microglobulin (*B2M*) was used as an endogenous control.

## Biofilm formation and metabolic activity

To assess pH-induced biofilm formation, *C. rodentium* was inoculated in LB at physiological pH to an $OD_{600}$ of 0.5 in a black, clear bottom 96-well plate. The plate was incubated at 37˚C, standing, for 48 h to allow for biofilm formation. After 48 h, culture media were removed and the $OD_{600}$ of the resultant biofilm was measured in a Synergy H1 plate reader (Biotek), and 100 µL of 3-(4,5-dimethylthiazol-2-yl)-2,5-diphenyltetrazolium bromide (MTT) at a concentration of 0.5 mg/mL in LB was added to the biofilms and allowed to incubate at 37˚C for 10 min. MTT was removed and replaced with 100 µL of 100% DMSO for resuspension before measuring OD at 570 nm to approximate formazan production as a readout of metabolic activity.

## Bacterial cell morphology

*C. rodentium* was cultured to mid-log phase (3.5 h, 37˚C with agitation) at gastrointestinal pH before being spun down and resuspended in PBS with the addition of FM 1–43 Dye (*N*-(3-Triethylammoniumpropyl)-4-(4-(Dibutylamino) Styryl) Pyridinium Dibromide) (Cat. no. T3163, Invitrogen). Images were taken on a Zeiss X10 light microscope and cell area was measured using CellProfiler 4.2.1 [37,38].

## Bacterial survival assays

To determine whether the deletion of *cpxRA* would affect *C. rodentium* survival at gastric pH, we cultured *C. rodentium* at pH 3.5 as described [23]. In brief, WT, Δ*cpxRA* [39], or Δ*cpxRA*:: *cpxRA C. rodentium* strains were inoculated 1:20 from overnight culture into unadjusted LB (the baseline "untreated" condition) or LB adjusted to gastric pH 3.5. At 30-min intervals, subcultures were sampled and diluted in PBS for plating on neutral LB agar to determine viability by quantifying surviving CFU. Viable CFU were compared to the inoculum population to determine percent survival over time.

To test for the acid tolerance response (ATR) following preinduction at intestinal pH, *C. rodentium* was cultured overnight at either neutral pH 7 or colonic pH 5.7 under anaerobic conditions (90% $N_2$, 5% $CO_2$, 5% $H_2$) or aerobically at 37°C without agitation. Overnight cultures were diluted 1:20 into unadjusted LB (the baseline "untreated" condition), or LB adjusted to gastric pH 3.5 and plated at 60-min intervals as previously described.

To determine whether *C. rodentium* shed from the mouse gut at early (day 2) and peak infection (day 7) differed in acid tolerance, mice were infected with *C. rodentium* as described above and fecal pellets were collected and homogenized fresh in 1 mL of PBS. Fecal homogenate was inoculated in LB pH 3.5 as described above and plated for enumeration of viable *C. rodentium* at 30-min intervals. A subset of fecal pellets was not homogenized to simulate natural encapsulation of *C. rodentium* during gastric acid exposure. Instead, fecal pellets were cut in half, with one half plated for CFU to determine the inoculating population, and the second half dropped into LB pH 3.5 as is for incubation as before. Pellets were removed from pH 3.5 after 120 min and homogenized for plating and CFU counts.

## Intra-bacterial redox potential

Assays were performed using a protocol adapted from [40,41]. In brief, a roGFP2 *C. rodentium* reporter strain was grown overnight in LB containing 100 μg/mL of carbenicillin. Overnight cultures were diluted 1:25 into 50 mL of pre-warmed LB without antibiotics and were grown for 5 h at 37°C in a shaking incubator. Bacterial strains were washed and resuspended in saline solution (0.9% NaCl w/w). Cultures were then exposed to the different pH of interest in a black 96-well plate with a volume of 200 μL and an $OD_{600}$ of 1. Intra-bacterial redox was assessed by measuring fluorescence with an excitation at 405 and 480 nm and an emission at 510 nm. Background fluorescence was controlled by measuring the fluorescence of WT *C. rodentium*. Normalized 405/480 nm ratios were calculated by using fully oxidized (100 mM $H_2O_2$) and reduced (10 mM DTT) control conditions.

## ELISA analysis of fasting serum gastrin levels

Blood was collected from naïve and day 8-infected mice following a 12-h fast. Whole blood was allowed to clot for 2 h on ice before serum collection. Gastrin levels were analyzed using the mouse gastrin enzyme-linked immunosorbent assay (ELISA) kit (Novus Biologicals, Cat. No. NBP3-08148). Absorbance at 450 nm was measured using a Synergy H1 plate reader (Biotek).

## ELISA analysis of inflammation

Serum calprotectin levels were analyzed using the mouse S100A9 ELISA kit (Hycult), as per the manufacturer's instructions. Serum was collected from non-fasted mice by allowing whole blood to clot for 2 h on ice before spinning to collect serum. Absorbance at 450 nm was measured using a Synergy H1 plate reader (Biotek).

## Cytokine measurements

To analyze intestinal inflammation of mice colonized with either *E. coli* Mt1B1, Δ*escN C. rodentium*, or WT *C. rodentium* as compared to naïve animals, colon tissue samples were collected at 4 days postinoculation by first removing intestinal contents. Samples were collected into 1 mL of PBS containing a protease inhibitor cocktail (Roche Diagnostics) and homogenized using a FastPrep-24 (MP Biomedicals) at 5.5 m/s for 2 min. Homogenate was then spun down at $16,000 \times g$ for 20 min at 4˚C for supernatant collection. Inflammatory cytokines were measured using a mouse inflammation cytometric bead array (CBA) kit (BD Biosciences, Cat. no. 552364) according to manufacturer's instructions. Measurements were made on an Attune NxT flow cytometer (Thermo Fisher Scientific), normalizing cytokine concentrations to tissue weight.

## Statistical analysis

Statistical analysis was performed in Graphpad Prism (www.graphpad.com) and clarified in figure legends. Unless otherwise stated, analysis was performed using a Mann–Whitney test to compare 2 groups and a one-way ANOVA with Šidák's multiple comparisons test for more than 2 groups. Aggregate results represent the mean +/− SD and statistical significance is represented by $^*p < 0.05$, $^{**}p < 0.01$, $^{***}p < 0.001$, and $^{****}p < 0.0001$.

# Results

## Regional variability in gastrointestinal pH may be influenced by genetic background

To assess the response of *C. rodentium* to intestinal pH, we first defined regional pH ranges within its natural host, the mouse. We compared reported gastrointestinal pH (pH 2 to 6) to pH measured in healthy, female, 7-week-old C57BL/6 mice across gut regions (Fig 1A and 1B). We observed a similar pattern to reported murine and human values, which see pH increase at significant increments towards the ileum-large intestinal regions (Fig 1B). At the cecum, we found a nonsignificant drop in pH which is characteristic of human intestinal pH, but not previously observed in mice [1]. Across all regions, we observed variability between individuals, particularly in the stomach and ileum, demonstrating that gastrointestinal pH likely fluctuates over time and with varied consumption of food and water.

Overall however, the pH across both the small intestine (duodenum, jejunum, and ileum) and large intestine (cecum and colon) was more basic than reported murine values in BALB/c mice [4]. Mice in the aforementioned study were fed a different diet, the Teklad Global 18% Protein Rodent Diet (Teklad), with a lower protein content (18%) [42]. Dietary protein is thought to increase intestinal pH by increasing ammonia production as a by-product of bacterial metabolism [2,9,43]. We therefore placed C57BL/6 mice on either the Teklad diet or Picolab Rodent Diet 20 5053 (Picolab; 21% protein) for 2 weeks before euthanasia. We found no significant differences between diets, except in the distal colon where pH was significantly higher on the lower protein Teklad diet (S1A and S1B Fig). This indicates that the more alkaline pH observed across gut regions may instead be due to differences in mouse genetic background (BALB/c versus C57BL/6). Indeed, while the only comprehensive profiling of murine intestinal pH has been done in BALB/c mice, ileal, cecal, and colonic pH has been reported to be more alkaline in FVB/N and ICR mouse backgrounds, and more closely representative of values found in the human gut (Fig 1) [44–46].

To verify whether mouse genetic background can indeed result in altered intestinal pH, we compared the intestinal pH of both BALB/c and C57BL/6 mice fed the Picolab diet. We found

a significant difference in intestinal pH in the ileum and colon regions (S1C Fig). As reported previously, we did not observe a drop in intestinal pH of BALB/c mice from the ileum to cecum regions, rather pH increased along the length of the GI tract. This indicates that genetic differences between mouse strains may result in altered intestinal pH, with pH of C57BL/6 mice more closely reflecting the pattern found in humans. Altogether, these data establish important physiological variation within the naïve murine gastrointestinal tract, which could be used by enteric pathogens to signal their biogeographical location.

## *C. rodentium* alters growth dynamics, epithelial attachment, and virulence gene expression in response to gastrointestinal pH

Having established the range of physiologically relevant pH, we next investigated whether *C. rodentium* responds to variations in pH by adjusting the pH of LB to mimic the ranges of pH found in both mouse and human GI tracts. Growth of *C. rodentium* was highest at the range of pH 6.4 to 7.5, with impaired growth at acidic pHs 4.4 and 4.8, resembling the gastric and duodenal regions (Fig 2A). As previously reported [23], no growth was observed at gastric pH of 3.5 to 4. Human enteropathogens EHEC, EPEC, and *S.* Typhimurium were similarly unable to grow at gastric pH 3.5 and displayed altered growth dynamics in response to pH below 5 (S2A–S2C Fig). This suggests that while acid tolerance and viability rates differ between pathogens [26,47], they are all affected by fluctuations in gastrointestinal pH.

We further investigated the relative metabolic activity of *C. rodentium* subjected to intestinal pH by measuring the reduction of MTT (3-(4,5-dimethylthiazol-2-yl)-2,5-diphenyltetrazolium bromide) to formazan by NADPH-dependent oxidoreductase enzymes in metabolically active bacteria. As expected, bacteria incubated at pH 3.5 produced significantly less formazan compared to neutral pH despite an increased biomass, suggesting decreased metabolic activity (S2D and S2E Fig). These results support both the absence of growth at pH 3.5 and the low recoverability of cells after removal from acidic pH [23]. The observed increase in biomass at pH 3.5 may further suggest the active encapsulation of community members within biofilms. Biofilm formation is an important stress-induced protective behavior of bacteria which allows for the survival of community members shielded within. Aside from direct death of bacteria under low pH, bacteria within biofilms are also known to decrease their metabolic activity under stress, which promotes survival and persistence, especially as many stressors affect the bacterial membrane (such as antibiotics) and have the greatest impacts on actively growing bacteria [48]. Taken together, extreme low pH of the gastric environment has considerable consequences to bacterial fitness.

To determine if the altered fitness dynamics observed across gastrointestinal pH also reflect differences in pathogen virulence, we assessed epithelial cell attachment, an essential characteristic of infection. This was done by "pre-inducing" *C. rodentium* via subculture in pH-adjusted LB before infection of CMT-93 cells, a murine rectal cell line chosen because the rectum is a known large intestinal niche of *C. rodentium* during host colonization. We expected attachment to increase in response to pH representing key replicative niches of *C. rodentium*, such as the cecum and colon. However, *C. rodentium* attachment was significantly increased after pre-induction at pH 3.5 and 4.4, representing gastric-duodenal pH (Fig 2B). Attachment was lowest at pH 7.5 which represents the upper range of ileal pH, as well as pH found in the extra-intestinal environment. Assays were repeated using subculture in pH-adjusted DMEM media, which is known to activate expression of the T3SS. Despite high buffering of DMEM media making accuracy difficult, the same trends were noted in both growth and attachment (S2F and S2G Fig). Therefore, LB media was used in all subsequent assays.

To further explore pH-mediated virulence regulation, we characterized the gene expression profile of known virulence- and early infection-associated genes following *C. rodentium*

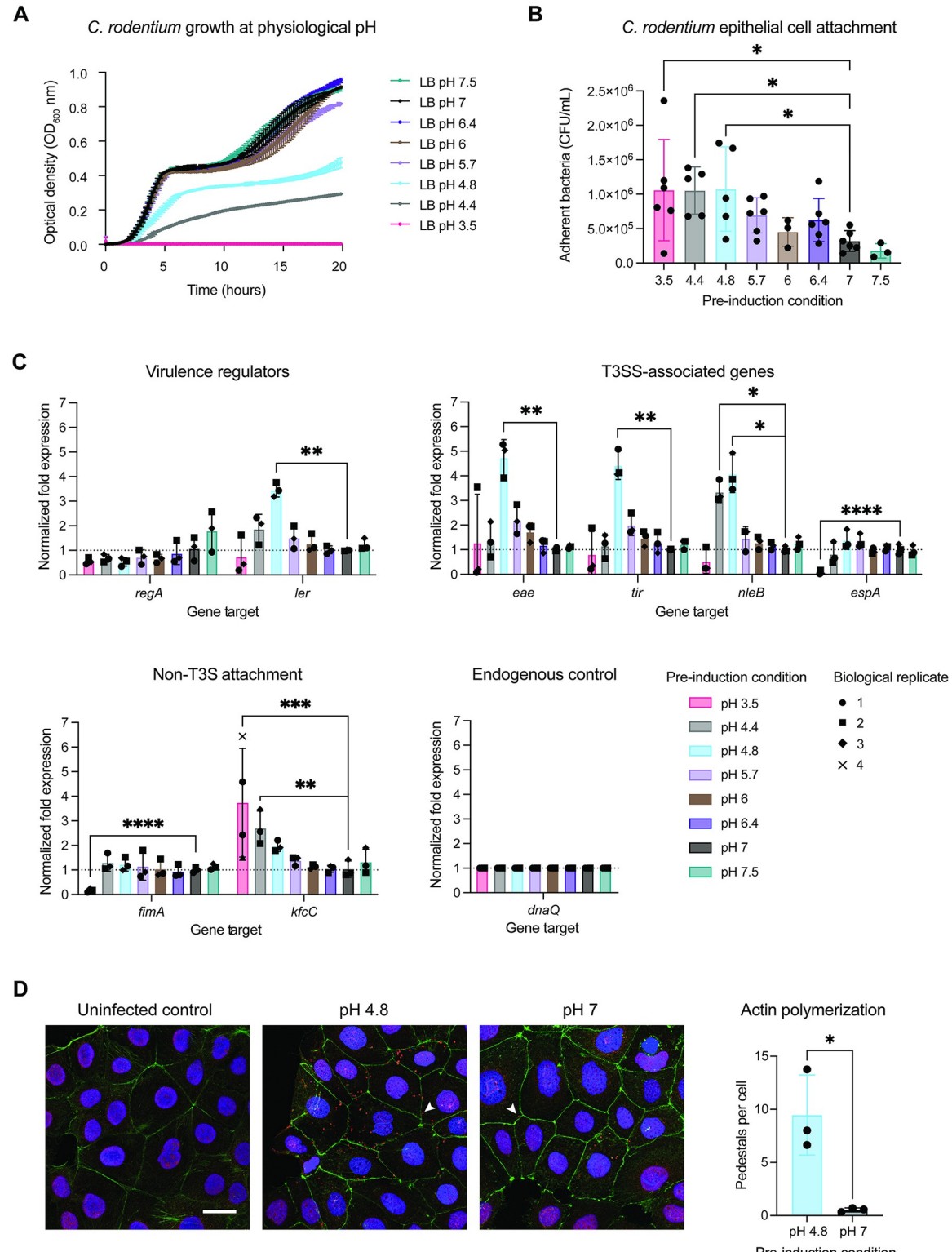

**Fig 2. *C. rodentium* alters its behavior and virulence in response to regional gastrointestinal pH.** (A) Growth in pH-adjusted LB across the range of gastrointestinal pH, measured as OD 600 nm at 10-min intervals over 20 h. (B) Attachment of *C. rodentium* to CMT-93 murine colonic epithelial cells after preinduction at gastrointestinal pH. Data represent biological replicates ($N$ = 3–4; each the average of 3 technical replicates). Statistical analysis represents an ordinary one-way ANOVA test with Holm–Šidák's multiple comparisons test. (C) Gene expression analysis by RT-qPCR of key virulence- and early infection-related genes across the range of

gastrointestinal pH. Statistical analysis represents *t* tests with Bonferroni correction for multiple comparisons. Outlier tests were run using the ROUT method (Q = 1%), identifying no outliers. Error represents mean +/− SD. (D) Representative images of actin pedestal formation on Caco-2-/TC7 cells following infection with *C. rodentium* pre-induced at pH 4.8 and pH 7.0 (*N* = 3 biological replicates). Scale bar represents 30 μm. Arrows indicate representative Tir-rich pedestals. Statistics represents an unpaired *t* test. Summary data displayed in Fig 2 can be found in S1 Data. LB, lysogeny broth; OD, optical density.

subculture in pH-adjusted LB prior to RNA extraction. Despite the fact that LB media is not known to activate the T3SS, type III secretion-associated virulence genes *ler*, *tir*, *eae*, and effector gene *nleB*, were highly expressed at pH 4.8 which represents duodenal pH (a 4- to 5-fold increase from baseline). At gastric pH 3.5, we observed down-regulation of virulence genes associated with type III secretion, including both structural and effector-associated genes (Fig 2C), as well as virulence regulators *ler* and *regA* [49]. While a known fimbrial gene *fimA* was also down-regulated, we found an up-regulation of *kfcC*, a component of the *C. rodentium* type IV pilus, which may explain the increase in epithelial cell attachment observed at gastric pH [49].

To further investigate the extent to which the T3SS contributes to pH-mediated attachment, we evaluated the ability of a Δ*escN C. rodentium* strain lacking the T3SS to attach to CMT-93 cells following preinduction as before. We found that infection with the Δ*escN* strain did not result in decreased epithelial attachment following preinduction in LB at low pH 3.5 and 4.4 (S2H Fig), further supporting that low pH induced epithelial attachment is mediated by non-T3SS factors. To determine whether the induction of T3SS-related genes at pH 4.8 results in increased actin polymerization and pedestal formation—a hallmark of infection by attaching and effacing pathogens such as *C. rodentium*, EHEC, and EPEC–CMT-93 cells were infected as before with either *C. rodentium* pre-induced at pH 4.8 or neutral pH 7.0. As CMT-93 cells do not demonstrate robust visual actin polymerization, we have further performed these assays using the human intestinal adenocarcinoma cell line Caco-2/TC7. Following infection, cells were stained for translocated intimin receptor (Tir), a core component of the T3SS used to stain for actin-rich T3SS-induced pedestals on epithelial cells [50]. We found that average number of pedestals per Caco-2/TC7 epithelial cell was significantly higher following 4-h infection with pH 4.8-induced *C. rodentium* (Fig 2D). As expected, *C. rodentium* was less able to induce pedestal formation in CMT-93 cells. However, we still observed a trend towards increased pedestal formation following infection with bacteria pre-induced at pH 4.8 (S2I Fig). Together, these data demonstrate that *C. rodentium* regulates virulence gene expression in a pH-dependent manner, leading to increased attachment-related behavior upon initial entry to the intestinal environment.

## Intestinal pH activates transcription of diverse bacterial stress responses

Bacterial stress responses and virulence are closely co-regulated [26,51]. Therefore, we next investigated the relationship between intestinal pH and the transcriptional expression of stress-related genes. A key strategy of food-borne pathogens under acid stress is to increase intracellular pH, which can be done via production of amino acid decarboxylases which allow for the consumption of cytoplasmic protons [52]. Interestingly, we did not find homology within the *C. rodentium* genome to either the transcriptional regulators GadW and GadX of the glutamate decarboxylase acid-resistance system, nor to AdiC of the arginine acid-resistance system found in some *E. coli* strains, including EHEC [53,54]. We did however identify 70% protein sequence homology between the *E. coli* CadA, a lysine decarboxylase, and *C. rodentium* LdcC (S3A Fig). As expected, we found that transcript levels of *ldcC* increased at low pH from 4.4 to 5.7 (Fig 3A), indicating a potential role for LdcC in survival of acid stress.

To further investigate this, we challenged wild-type *C. rodentium* and a *C. rodentium* strain lacking *ldcC* to gastric pH 3.5 over a range of 0 to 120 min. We found that viability of the Δ*ldcC* strain was significantly impaired at 120 min as compared to the wild-type strain (S3B Fig). These data indicate that *C. rodentium* uses amino acid decarboxylation to combat acid-induced stress despite lacking homology to 2 major systems in related *E. coli*. However, the effect of *ldcC* deletion was subtle, with no significant difference observed upon initial exposure to low pH, indicating that *C. rodentium* likely employs several strategies of acid adaptation.

Another strategy of food-borne pathogens in dealing with acid stress is proton efflux [52]. The $F_1F_0$-ATPase proton efflux membrane ATPase is conserved across many bacterial pathogens, including *C. rodentium*, in which it is encoded by the *atpD* gene. We observed a 7-fold increase in expression of *atpD* at pH 4.4 compared to neutral media (Fig 3A). Additionally, upon searching the *C. rodentium* genome for genes annotated as involved in acid resistance, we chose to further investigate the expression of *yodD*, a stress-induced peroxide/acid resistance protein, and *asr*, the acid resistance repetitive basic protein, both of which are shared with EHEC. Both genes were up-regulated at gastric pH 3.5, and *asr* in particular demonstrated a 300-fold increase in expression at pH 4.4, suggesting that both of these genes are important to *Citrobacter*'s acid resistance strategy within the upper GI tract.

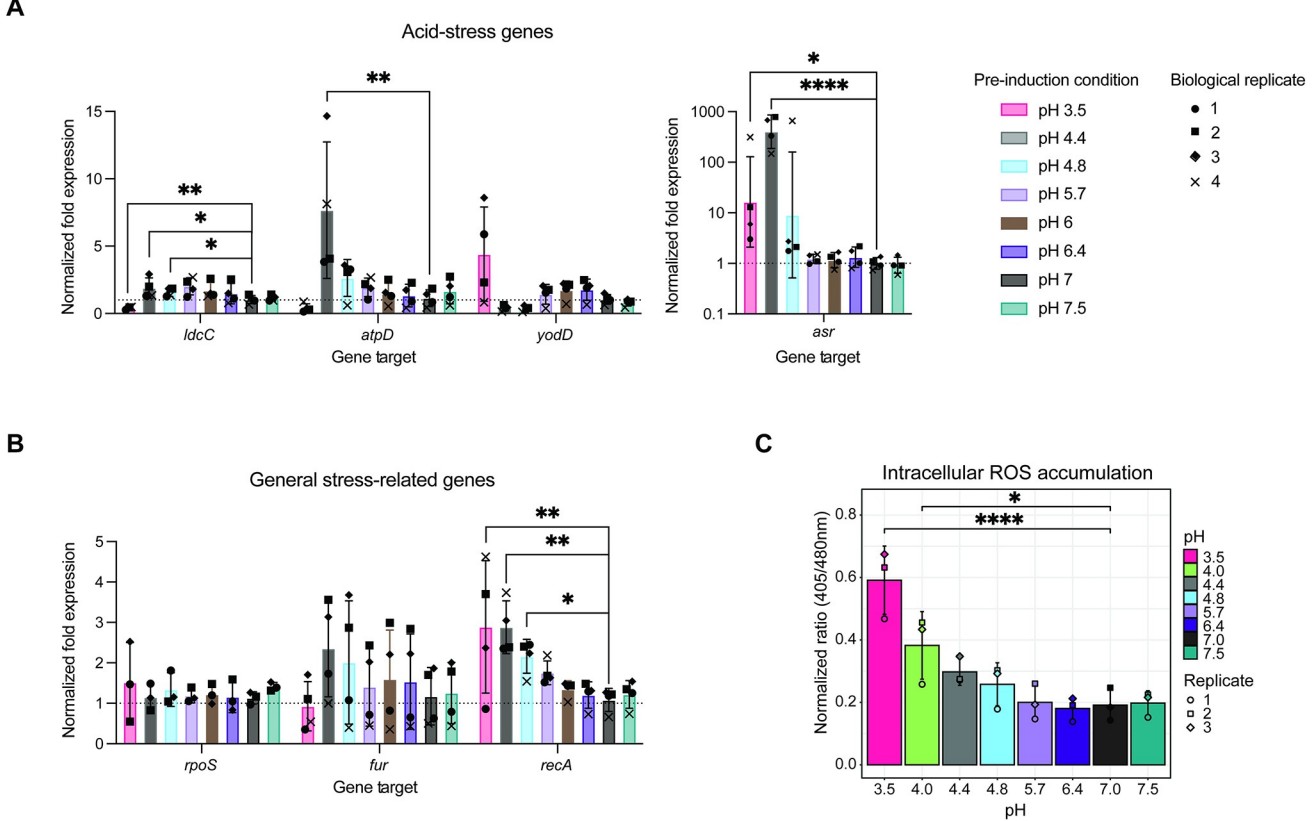

**Fig 3. Stress gene regulation is affected by gastrointestinal pH.** (A) Expression of acid stress-associated genes across the range of gastrointestinal pH by RT-qPCR. Data represent biological replicates (average of 3 technical replicates; $N = 4$). (B) Expression of general stress-associated genes across the range of gastrointestinal pH by RT-qPCR. Data represent biological replicates (average of 3 technical replicates; $N = 3–4$). (C) Intracellular ROS accumulation at 3 h postinoculation into LB at gastrointestinal pH, as determined using a roGFP2 reporter *C. rodentium* strain. Data represent biological replicates (average of 3 technical replicates; $N = 3$). Statistical analysis represents *t* tests with Bonferroni correction. Outlier tests were run using the ROUT method (Q = 1%), identifying no outliers. Summary data displayed in Fig 3 can be found in S1 Data. LB, lysogeny broth; ROS, reactive oxygen species.

Aside from direct mechanisms of acid resistance, we further profiled the effect of intestinal pH on nonspecific stress responses which have evolved as general mechanisms to overcome environmental stress. We investigated the expression of *rpoS*, a gene encoding the stationary phase sigma factor present in Gram-negative bacteria which controls general stress responses to stressors such as oxidative or osmotic stress. RpoS is important for *C. rodentium* survival under heat and $H_2O_2$ conditions, as well as full pathogen virulence [55,56]. Mutation of *rpoS* in EHEC has also been found to decrease resistance to acid stress [57]. While *rpoS* expression was therefore expected to increase with decreased pH, it was unaltered across the pH range tested (Fig 3B). This was surprising, given that when we exposed a roGFP2 reporter strain of *C. rodentium*, which can be used to measure intra-bacterial redox dynamics, to different pHs we found that gastric pH 3.5 to 4 significantly increased oxidative stress (Fig 3C) [41]. We further investigated expression of *fur*, encoding a ferric iron transcription regulator which may regulate transcription of genes that protect against ROS damage [52]. While we observed a trend towards increased *fur* expression at pH 4.4, there was no significant up- or down-regulation of the system induced by altered pH. Despite the lack of ROS-specific gene expression, we did observe evidence that bacteria at low pH respond to general cell damage. For example, we found that expression of *recA* which encodes a recombinase protein used to repair DNA double-strand breaks increased with increased acidity of the culture media, suggesting cell stress and DNA damage at pH 4.8 and lower (Fig 3B) [58].

## pH-induced stress alters membrane regulation and cellular morphology

An important outcome of stress-induced gene regulation is the repair and maintenance of the bacterial cell membrane, which can even induce adaptive changes to cellular morphology. To visualize potential morphological changes induced by pH, *C. rodentium* was pre-induced at GI pH before staining with FM 1–43 Dye. We observed bacterial cell lengthening at pH 4.0 to 4.8, an indicator of cell stress, and a trend towards small size and abnormal cell shape at pH 3.5 (Fig 4A and 4B). We also observed a decrease in membrane intensity in a subpopulation of cells at pH 3.5, possibly reflecting a loss of membrane integrity [23].

The Cpx envelope stress response is an important mechanism to overcome conditions which result in misfolding of envelope proteins, such as alkaline pH, and has been shown to be required for *C. rodentium* virulence [51,59]. We found that *cpxA*, part of the CpxRA two-component system which mediates the Cpx response, was most highly expressed at low pH 4.4 [51]. CpxRA is known to regulate both the expression of periplasmic proteases such as DegP, as well as T3SS-related virulence. We found that expression of *degP*, which is induced by stress [60], was significantly up-regulated in response to subculture at pH 3.5, 4.4, 5.7, and 6.0 compared to neutral base media (Fig 4C). *degP* was not up-regulated at pH 4.8 which resulted in a strong activation of T3SS-related genes, consistent with reported inverse regulation of these systems (Fig 2C), and supporting Cpx stress response regulation under low upper intestinal pH conditions.

The above data suggest that membrane protein synthesis and degradation must be tightly regulated to allow for successful pathogen adaptation to variable intestinal pH. If this is the case, interfering with low pH-induced membrane regulation will impact *C. rodentium* survival under acidic conditions. We therefore tested survival at gastric pH 3.5 of wild-type *C. rodentium* or a strain deficient in the CpxRA two-component system ($\Delta cpxRA$), and which is unable to properly regulate its membrane stress response [51]. While there was no difference upon initial exposure to pH 3.5 (0 min) between wild-type, $\Delta cpxRA$, and $\Delta cpxRA$::*cpxRA* strains, the $\Delta cpxRA$ strain exhibited significantly reduced survival after 30 to 120 min, with up to an average of 3.4% viability in the absence of *cpxRA* after 120 min, which represents the upper limit of

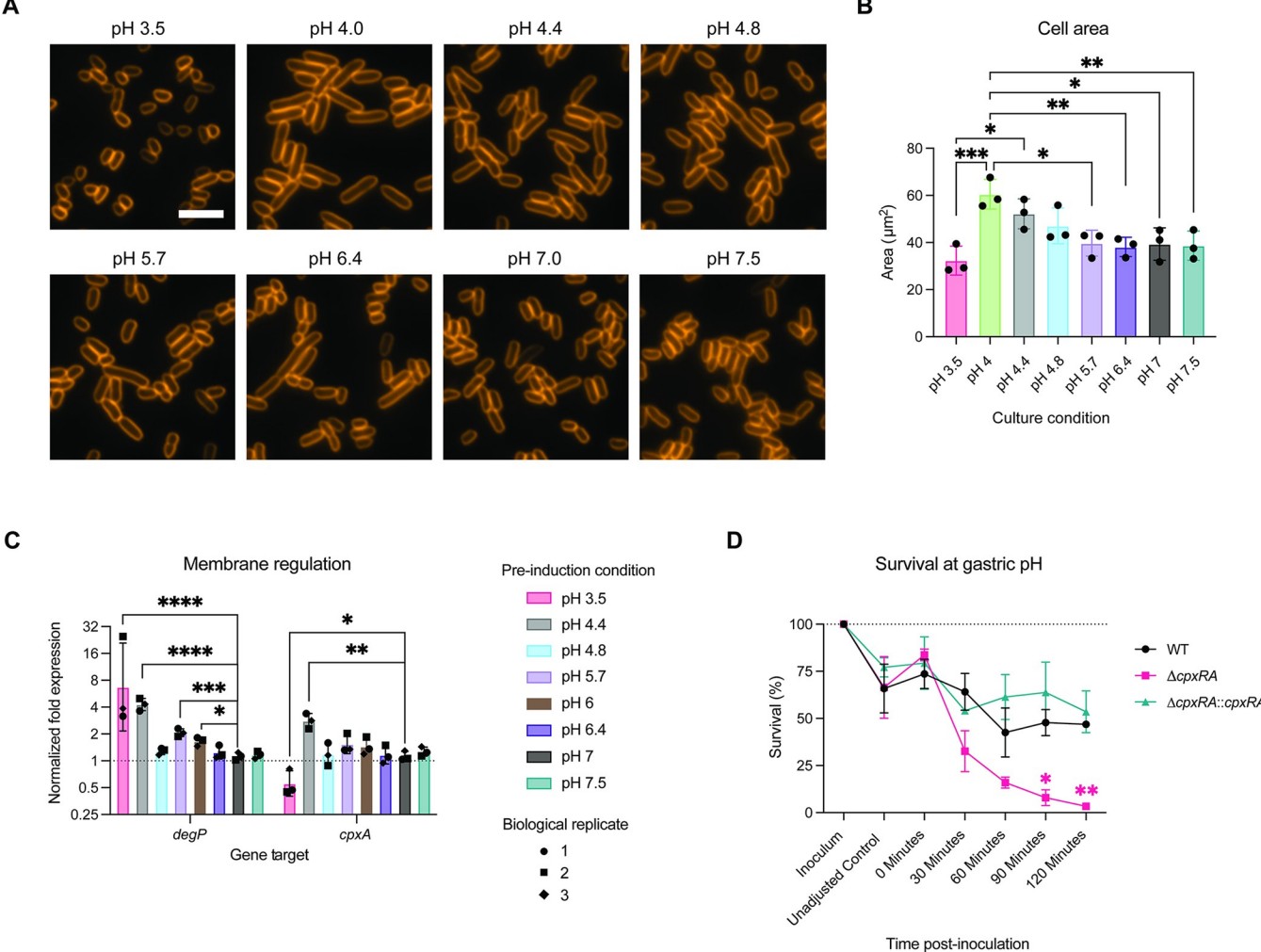

**Fig 4. Membrane regulation in response to intestinal pH.** (A) Representative cellular morphology after 3.5 h at relevant gastrointestinal pHs by light microscopy and cell membrane staining. Scale bar represents 5 μm. (B) Cell area measurements from light microscopy across gastrointestinal pH (*N* = 3 biological replicates, each the sum of 2 technical replicates). (C) Expression of genes by RT-qPCR associated with the Cpx stress response following culture at gastrointestinal pH. Statistical analysis represents *t* tests with Bonferroni correction for multiple comparisons. Outlier tests were run using the ROUT method (*Q* = 1%), identifying no outliers. Data represent biological replicates (average of 3 technical replicates; *N* = 3). (D) Survival of wild-type, Δ*cpxRA*, Δ*cpxRA*:: *cpxRA C. rodentium* in gastric pH 3.5 over the range of average gastric emptying time. Each point represents the average of 3–5 biological replicates (3 technical replicates per biological replicate). Statistical analysis represents a mixed-effects model with Geisser–Greenhouse correction and Šidák's multiple comparisons test. Color of stars indicates significance of Δ*cpx* (pink) or complemented (green) strains compared to the WT strain. Summary data displayed in Fig 4 can be found in S1 Data.

gastric emptying time, as compared to 53.5% in the complement strain (Fig 4D) [61]. The Cpx response is therefore not only necessary for *C. rodentium* virulence, but is also critical to gastric passage and overall host–host transmission. Overall, these data suggest that *C. rodentium* alters its membrane and overall shape under gastrointestinal pH conditions to adapt to its surrounding environment.

## Host-adaptation increases *C. rodentium* acid tolerance which may support host transmission

During an established infection, *C. rodentium* shed from the gut at peak infection is known to display a hypervirulent state characterized by altered T3SS expression and tissue colonization

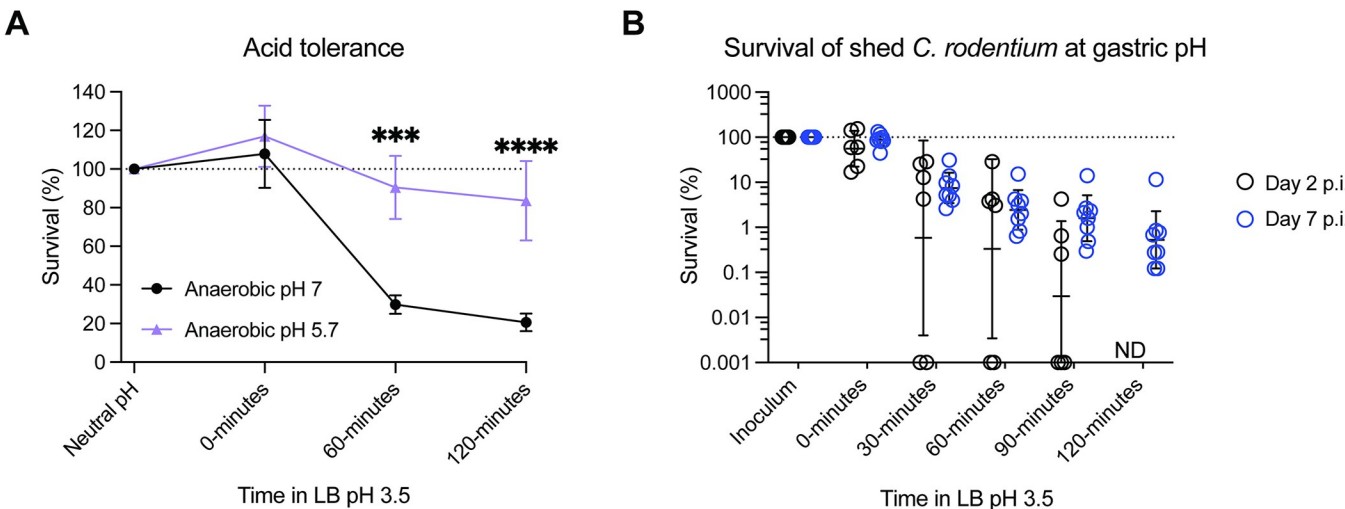

**Fig 5. HA *C. rodentium* is better able to survive acid-induced stress.** (A) *C. rodentium* survival at gastric pH after initial anaerobic preinduction at either colonic pH 5.7 or neutral pH. Each point represents the average of 3 biological replicates (3 technical replicates per biological replicate). Statistics represent a two-way ANOVA with Šidák's multiple comparisons test. (B) Survival of *C. rodentium* in homogenized feces collected on days 2 and 7 postinfection, representing non-adapted and host-adapted states (*N* = 6–8). ND, not detected. Summary data displayed in Fig 5 can be found in S1 Data. HA, host-adapted; LB, lysogeny broth.

dynamics, and the ability to colonize new hosts at a reduced infectious dose, allowing for more effective transmission. This hypervirulent state is quickly lost following *ex vivo* passage, indicating that it results from within-host adaptation rather than evolutionary change [62,63]. We hypothesized that a crucial part of this host-adapted (HA) state would include adaptation to better tolerate the gastric environment. While the conditions that trigger host-adaptation remain unknown, since host-adaptation occurs following passage through the mouse gut, we investigated whether *in vitro* culture at colonic pH representing the key *in vivo* niche of *C. rodentium*, ahead of gastric pH exposure, could induce an ATR and mimic the protection observed after gastrointestinal transit. Indeed, we observed a striking ATR following stationary phase growth at colonic pH 5.7 under anaerobic conditions (Fig 5A). While culture at neutral pH resulted in only 20.6% survival after 2 h at gastric pH, subculture at colonic pH 5.7 resulted in an average survival rate of 83.6%. Anaerobic conditions were chosen to mimic passage through the low oxygen large intestinal environment. We did not observe an ATR under aerobic conditions nor in response to short-term exposure to low pH (preinduction to mid-log phase) (S4A Fig).

Having demonstrated a robust ATR under colonic pH conditions, we next sought to determine whether HA-*C. rodentium* conditioned within the host gut, demonstrates increased acid tolerance. We evaluated the survival of *C. rodentium* shed in the feces of infected mice at day 2, representing an early, non-HA state, and *C. rodentium* shed at day 7 postinfection, representing the HA state (Fig 5B). Though we found similar percent survival at 0 to 30 min postexposure to pH 3.5, none of the day 2-shed *C. rodentium* was able to survive 120 min at gastric pH. As these fecal samples were homogenized before exposure to pH 3.5 rather than naturally encapsulated in the fecal pellet, we repeated this assay on day 2 feces without homogenization. Despite an increase to 8% survival, we still observed significant population loss (S4B Fig). This indicates that while natural encapsulation, such as within contaminated food, contributes to "naïve" non-HA pathogen passage of the gastric barrier it must be combined with acid adaptation to facilitate high efficiency transmission. Altogether, these data suggest that *C. rodentium* adapts to the acidic conditions of the colon during colonization, possibly conferring a competitive advantage to initial gut seeding.

## Changes to host cell regulation and circulating gastrin levels result in decreased gastric pH following *C. rodentium* infection

All the above experiments were done by subjecting *C. rodentium* to the range of intestinal pH found within the naïve murine gut. However, many global gut environmental changes occur during enteric infection. Therefore, we next sought to profile intestinal pH at peak *C. rodentium* infection to determine if infection influences environmental pH, either as a result of pathogen colonization or an active host response. To test this, we measured gastric and cecal pH of infected mice at an early (day 4) and peak (day 8) infection time point and compared values to the corresponding pH in uninfected control mice. Cecal pH did not change at days 4 or 8 postinfection (pi), despite a high pathogen burden (Fig 6A). However, we found that gastric pH significantly decreased by day 8 pi as compared to uninfected mice, correlating with colonic pathogen burden (Fig 6A and 6B). This was not due to decreased food consumption by the mice (S5A and S5B Fig) [4].

To determine whether this decrease in gastric pH was the result of altered acid secretion, we analyzed host gene expression within the stomach tissue of naïve and infected mice at day 8 pi. We observed a nonsignificant increase in *ATP4B* which encodes the beta subunit of the gastric H+,K+-ATPase, a proton pump responsible for acid secretion in parietal cells (Fig 6C) [64]. We then specifically investigated changes across histamine-, gastrin-, and acetylcholine-mediated pathways of gastric cell activation. We observed a significant increase in *HRH2* expression, encoding histamine receptor H2, a G protein-coupled receptor that stimulates gastric acid secretion in response to histamine (Fig 6C) [64–66]. Additional genes associated with histamine-mediated acid secretion, such as *HDC* (encoding the histidine decarboxylase which converts L-histidine to histamine), and *PAC1* (encoding a receptor involved in histamine release from gastric enterochromaffin-like cells) remained unaltered (Figs 6C and S5C).

Hormonal activation of gastric acid secretion is facilitated by release of gastrin, a hormone produced by gastric G cells and released into the circulation following food consumption [64–66]. We observed a nonsignificant trend towards increased *GAST* gene expression, which directly encodes gastrin (Fig 6C). Expression of *CCK2R*, encoding the cholecystokinin B receptor which responds to gastrin, was highly variable in expression among infected mice (S5C Fig). Furthermore, although cholinergic agents also facilitate parietal cell activation, we did not observe changes to the expression of *CHRM3*, a gene encoding the M3 muscarinic acetylcholine receptor which responds to the release of cholinergic agents (S5C Fig).

We also investigated genes related to negative control of gastric acid secretion, specifically gastric levels of *VIP* gene expression encoding vasoactive intestinal peptide and the gene encoding its associated receptor in the stomach, *VPAC2* [64–66]. We found a nonsignificant trend towards decreased levels of *VPAC2* (S5D Fig; $p = 0.08$), indicating that infection potentially leads to dysregulation of multiple systems involved in maintaining gastric pH. Collectively, these data suggest changes to parietal cell activation postinfection.

To assess whether transcript-level expression reflects systemic changes to regulation of gastric pH, we measured fasting serum gastrin levels by ELISA. High levels of circulating gastrin detected in a fasted state are indicative of hypergastrinemia, and gastrin levels are thereby used as a diagnostic marker in humans [65]. Indeed, we found that circulating gastrin levels in *C. rodentium*-infected mice at day 8 pi were significantly higher than in naïve mice (Fig 6D). Taken together, our data support decreased gastric pH post-*C. rodentium* infection, resulting from active changes to host acid regulation.

## Gastric pH decreases in response to further infection and chemical models of epithelial barrier dysfunction

To further investigate decreased gastric pH as an active host defense mechanism, we next wished to determine whether similar changes could be observed in other enteric infection

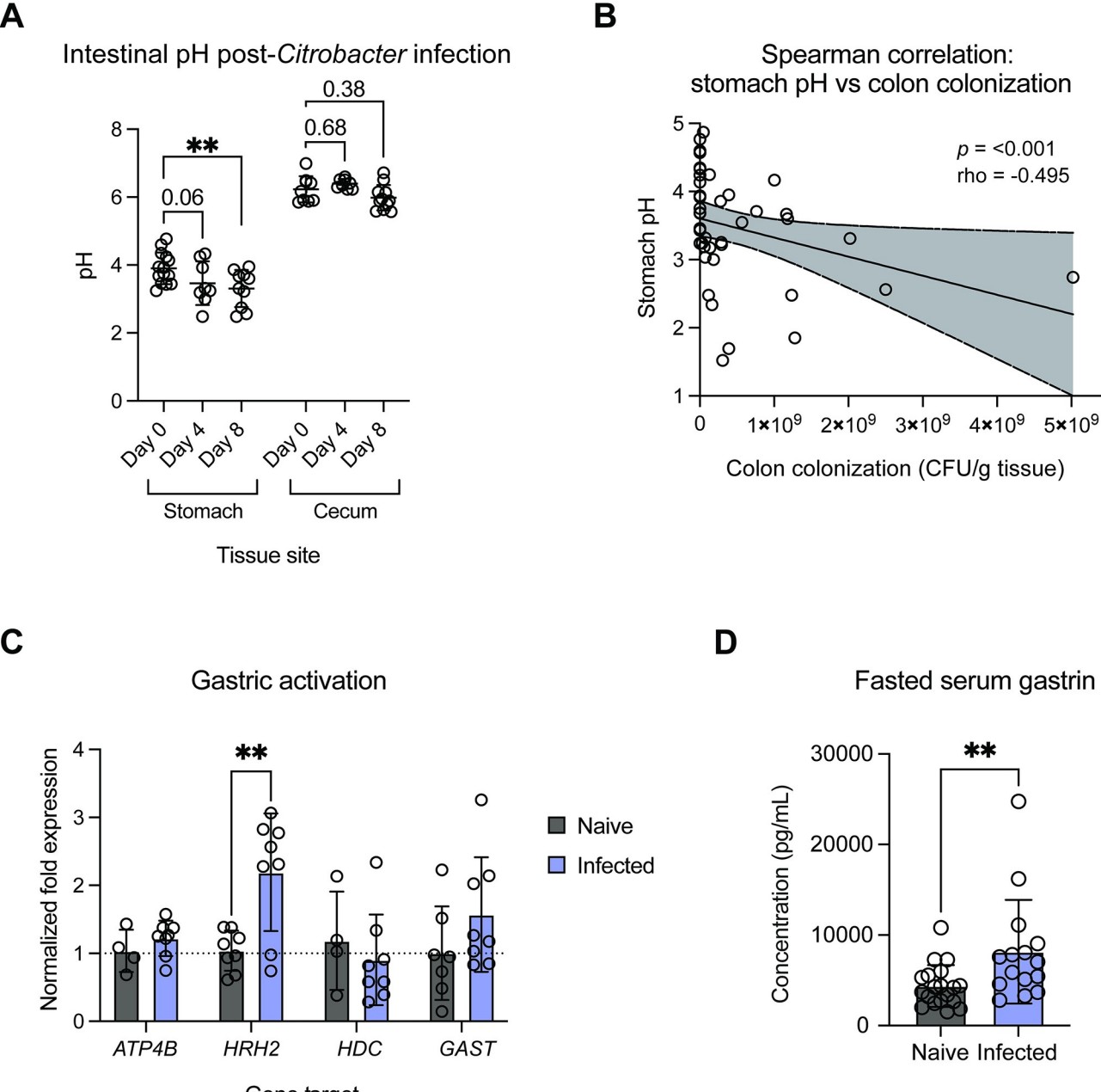

**Fig 6. Murine gastric pH decreases at peak infection.** (A) Intestinal pH measurements in the murine stomach and cecum over time post-*C. rodentium* infection (inoculum dose of $10^8$ CFU). Statistical analysis represents a two-way ANOVA with Dunnett's multiple comparisons test ($N = 8$–14). (B) Spearman correlation of stomach content pH and colon colonization (CFU/gram tissue) at days 4 and 8 postinfection. Line represents linear regression +/− 95% CI. *P*- and rho-values represent Spearman correlation values. (C) Gene expression analysis of host genes associated with stomach parietal cell activation and acid secretion within stomach tissue from naïve and infected mice ($N = 4$–8). Statistics represent a two-way ANOVA with Šidák's multiple comparisons test. (D) Fasted serum gastrin levels in naïve and day 8-infected mice. Statistical analysis represents an unpaired *t* test ($N = 15$–19). Summary data displayed in Fig 6 can be found in S1 Data. CFU, colony-forming unit.

models. To address this, we infected mice with human pathogen *Salmonella enterica* serovar Typhimurium and measured gastric pH at peak infection on day 4 pi. We observed a significant decrease in gastric pH in response to *S.* Typhimurium (Fig 7A) which significantly correlated with pathogen burden (S6A Fig), demonstrating consistency across additional enteric

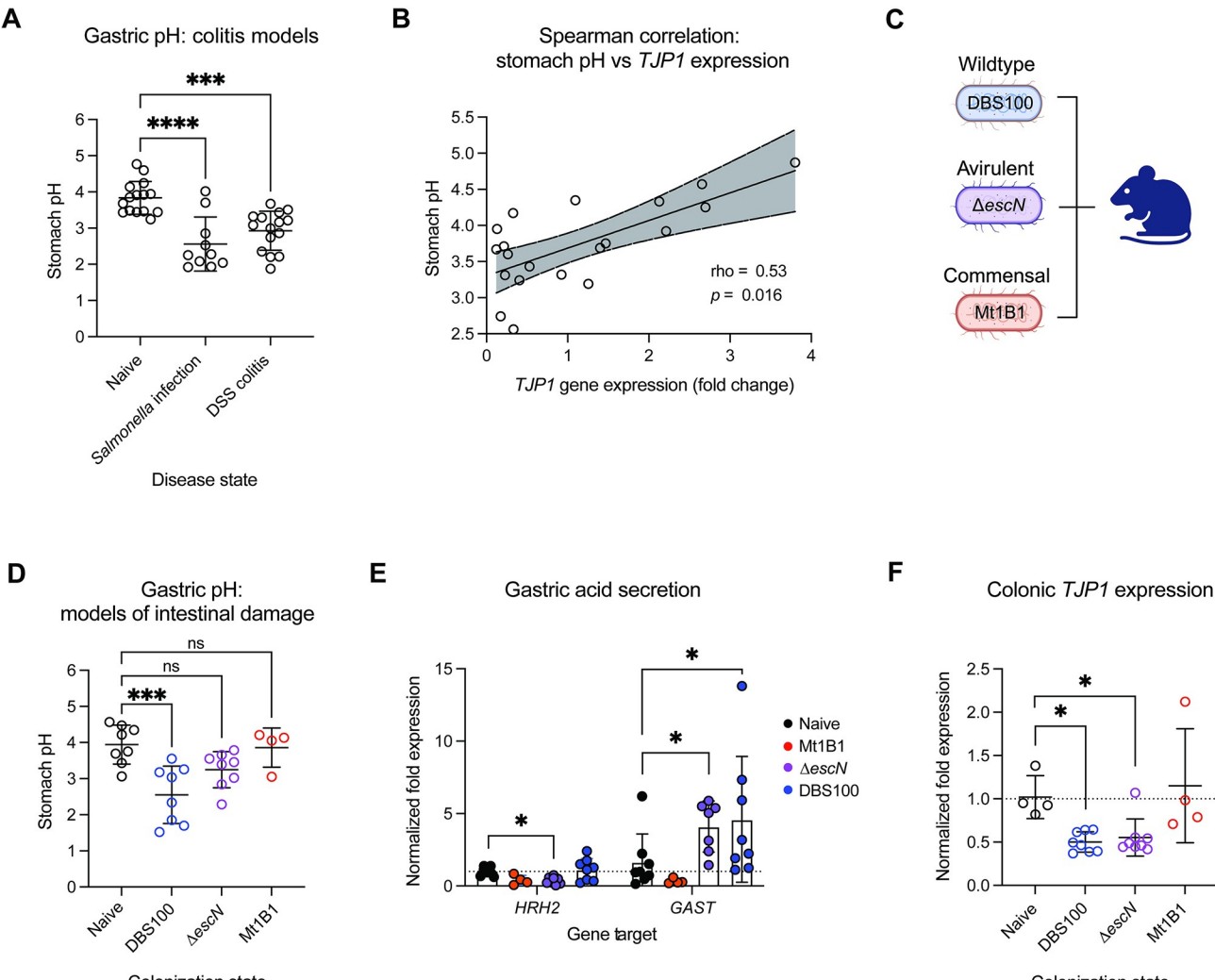

**Fig 7. Murine gastric pH decreases in additional models of intestinal damage.** (A) Measurements of gastric pH in additional models of intestinal damage: *Salmonella enterica* serovar Typhimurium typhoid model of infection (*Salmonella* infection), and the DSS model of colitis (DSS colitis) ($N = 10$–15). Statistical analysis represents an ordinary one-way ANOVA with Tukey's multiple comparisons test. (B) Spearman correlation of stomach pH and *TJP1* expression within colonic tissues of naïve and infected mice ($N = 20$). Line represents linear regression +/− 95% CI. *P*- and rho-values represent Spearman correlation values. (C) Summary of strains used to test the impact of intestinal damage on gastric pH: wild-type *C. rodentium* DBS100, T3SS-deficient Δ*escN C. rodentium*, and commensal *E. coli* Mt1B1. (D) Measurement of gastric pH in mice which are uninfected, or colonized with either wild-type *C. rodentium*, Δ*escN C. rodentium*, or commensal isolate *E. coli* Mt1B1 at an inoculum dose of $2$–$4 \times 10^9$ CFU. Measurements were taken at euthanasia at day 4 postinoculation ($N = 4$–8). Statistical analysis represents an ordinary one-way ANOVA with Bonferroni's multiple comparisons test. (E) Gene expression analysis of host genes *HRH2* and *GAST* within stomach tissue from naïve mice and mice colonized with either wild-type *C. rodentium*, Δ*escN C. rodentium*, or commensal isolate *E. coli* Mt1B1 at day 4 postinoculation ($N = 4$–8). Statistics represent a two-way ANOVA with Šidák's multiple comparisons test. (F) Colonic *TJP1* expression in naïve mice and mice colonized with either wild-type *C. rodentium*, Δ*escN C. rodentium*, or commensal isolate *E. coli* Mt1B1 at day 4 postinoculation ($N = 4$–8). Statistics represent a Kruskal–Wallis test with Dunn's multiple comparisons test. Summary data displayed in Fig 7 can be found in S1 Data. Created with BioRender.com. CFU, colony-forming unit; DSS, dextran sulfate sodium; T3SS, type III secretion system.

infection models. To determine whether this response is infection specific, we next performed a DSS model of chemically induced colitis. Despite no difference between the pH of control and DSS drinking water (S6B Fig), we again observed a significant decrease in gastric pH following 4 days of chemical administration (Fig 7A). We therefore hypothesized that gastric pH may decrease in response to intestinal damage rather than infection specifically.

To investigate the relationship between intestinal damage and gastric pH, we measured colonic expression of tight junction protein TJP1, a marker of epithelial barrier integrity. Indeed, we observed a positive correlation between *TJP1* expression and gastric pH during *C. rodentium* infection (rho = 0.53, *p* = 0.02; Figs 7B and S6C), such that lower gastric pH reflected decreased colonic barrier integrity. In contrast, serum levels of calprotectin, a general marker of systemic host inflammatory state altered by infection, did not correlate with gastric pH (rho = −0.299, *p* = 0.2; S6D and S6E Fig).

To further test the relationship between intestinal damage and gastric pH, we colonized mice with either wild-type *C. rodentium*, a type III secretion system-deficient mutant strain (Δ*escN*), or murine commensal *E. coli* isolate Mt1B1 (Fig 7C). The Δ*escN* and commensal Mt1B1 strains are capable of causing only a moderate to negligible level of tissue pathology and inflammation upon murine colonization, respectively [35,36,67], as confirmed by inflammatory cytokine levels in the colon (S6F Fig). Due to the inability of the Δ*escN* to persist within the mouse gut, we sampled mice at 4 days pi when bacterial burdens were expected to be similar across all groups. We found that gastric pH did not decrease in response to commensal *E. coli* Mt1B1, and that Δ*escN* inoculation resulted in only a mild, nonsignificant, decrease in gastric pH (Fig 7D). Furthermore, unlike in wild-type *C. rodentium* infection, colonization rates of Δ*escN* DBS100 and Mt1B1 did not correlate with gastric pH (S6G Fig), suggesting that bacterial colonization in the absence of tissue pathology does not affect gastric pH.

We further measured expression levels of *HRH2*, which were found to be significantly lower in response to colonization by the Δ*escN* strain, with no differences observed between naïve and Mt1B1-colonized mice (Fig 7E). However, we did find elevated expression of *GAST* in Δ*escN* colonized mice, likely accounting for the moderate decrease observed in gastric pH (Fig 7E). This increase in *GAST* expression again corresponded with decreased expression of *TJP1* in the colon of mice colonized with the Δ*escN* strain but not *E. coli* Mt1B1, further supporting the relationship between colonic epithelial barrier integrity and gastric pH (Fig 7F). Together, these results indicate that decreased gastric pH may be a widespread host response to intestinal damage.

## Discussion

Gastrointestinal pH varies along both the human and mouse gut, likely serving as an important indicator of intestinal geography for invading pathogens. Our study reveals that enteric pathogen *C. rodentium* responds to subtle pH changes representative of gut pH (even changes of <0.5), altering its growth and virulence. Exposure to pH resembling the cecal environment, *C. rodentium*'s optimal niche, promoted pathogen growth but did not stimulate the expected T3SS-related attachment. Instead, we found that small intestinal pH 4.4 to 5.7 increased expression of T3SS genes. Duodenal pH may signal pathogen entry to the small intestine from the acidic stomach environment, a transition that necessitates attachment to the host epithelium to prevent pathogen expulsion from the GI tract. Interestingly, gastric pH also enhanced epithelial cell attachment, this time through up-regulation of the type IV pilus Kfc (K99 fimbrial-like adhesin), an early attachment factor, rather than the T3SS (Fig 2). Collectively, our data suggest that low pH is a trigger of loose intestinal attachment upon immediate entry to the small intestine, while the higher pH of the large bowel favors pathogen expansion to support a high pathogen load [23].

Unexpectedly, we found that gastric pH was significantly decreased at peak infection (Fig 6). Up-regulation of host genes associated with acid secretion suggests that stomach acidification is an active adaptation of the host. Histamine receptor-2 (HRH2) in particular is associated with gastric ulcers such that $HR_{H2}$-blockers, such as rantidine (Zantac), are used to treat

conditions involving excessive stomach acid production [68,69]. We found that this response may not be pathogen specific, but instead related to intestinal damage, demonstrating the same phenotype during chemically induced colitis but not in response to colonization by commensal *E. coli* nor by Δ*escN C. rodentium* lacking key damage-inducing virulence machinery (Fig 7). The host environment has evolved many adaptive responses to clear invading bacteria such as fever, vomiting, and both specific and nonspecific immune responses [29,70]. Decreasing stomach pH may be another important host response for protection against further pathogen ingestion. Surprisingly, downstream cecal pH did not change at peak infection despite high pathogen burdens which were expected to acidify the gut content (Fig 6). Enteric pathogen *Clostridium difficile* has been shown to actively inhibit ion exchange to promote pathogen-favorable conditions, resulting in more alkaline stool in infected patients [71]. It may be that *C. rodentium* is similarly able to manipulate ion transport at key niche locations, such as in the cecum, to maintain favorable conditions for pathogen expansion.

While a decrease from an average gastric pH of 4.0 to 3.5 during peak *C. rodentium* infection may seem inconsequential, we observed significant differences in cellular morphology and metabolic activity between pHs. While pH 3.5 induced small cell size and decreased metabolic activity, a pH increase of only 0.5 resulted in cell elongation and metabolic recovery, and a reduction in intracellular ROS (Figs 3, 4, and S2). Therefore, even incremental host changes may present a formidable barrier to intestinal re-entry. Interestingly, "host-adapted" *C. rodentium* shed at peak infection demonstrated higher acid tolerance than host-naïve *C. rodentium*, indicating that the pathogen develops a more robust ATR during passage through the mouse gut, which may be triggered by exposure to colonic pH conditions (Fig 5). Acid adaptation may thereby contribute to the enhanced transmission of host-adapted *C. rodentium* as compared to host-naïve *C. rodentium* [62]. Taken together, it is clear that adaptive pH responses occur on both sides to maintain the balance between pathogen and host.

Given the impact of enteric infection on gastric acid secretion in mice, intestinal damage and inflammatory state may have important functional consequences to gut pH homeostasis. Few studies investigate human gastrointestinal pH following infection or other stressors. However, a known association exists between functional dyspepsia (FD) and infectious gastrointestinal disease by bacteria, viruses, and protozoa [72]. FD is a disorder of unknown etiology characterized by stomach pain and other indigestive symptoms, which occasionally overlaps with postinfectious irritable bowel syndrome [73,74]. A meta-analysis of 17 clinical studies found onset of postinfectious FD in 9.5% of adult subjects, indicating a 2.5-fold risk of FD following acute gastroenteritis [72]. Therefore, further understanding of gastrointestinal pH in response to environmental triggers or altered host health, such as infectious disease state, could be important to predicting susceptibility to secondary intestinal disease.

In summary, our study illustrates how regional GI fluctuations in pH signal changes to *C. rodentium* behavior and virulence, with consequences to bacterial stress and survival. We further observed infection-induced alterations to host pH which could impact the bioavailability and effectiveness of pH-dependent release drugs, or colonization of probiotic bacterial strains for the treatment of intestinal morbidities. As such, there is immense value in understanding the complex and long-lasting interactions of pathogens and their hosts.

## Supporting information

**S1 Fig. Effect of standard mouse chow on intestinal pH of C57BL/6 mice.** (A) Murine intestinal pH after 2 weeks on either standard Teklad 18% protein mouse chow or Picolab 21% protein mouse chow. Statistical analysis represents a mixed-effects model with Geisser–Greenhouse correction for matched values and Tukey's multiple comparisons test (*N* = 6). (B)

pH of Teklad 18% protein standard mouse chow compared to Picolab 21% protein standard mouse chow. Statistical analysis represents an unpaired *t* test (*N* = 3 pellets, average of 3 pH readings). (C) Murine intestinal pH of C57BL/6 and BALB/c mice after 2 weeks on Picolab 21% protein mouse chow. Statistical analysis represents a mixed-effects model with Geisser–Greenhouse correction for matched values and Tukey's multiple comparisons test (*N* = 8–18). Summary data displayed in S1 Fig can be found in S1 Data.
(TIF)

**S2 Fig. Similar patterns of growth observed in human pathogens EHEC, EPEC, and *S.* Typhimurium at physiological pH.** Growth of (A) enterohemorrhagic *E. coli* (EHEC), (B) enteropathogenic *E. coli* (EPEC), and (C) *S.* Typhimurium at physiological pH over 20 h, indicating that all 3 pathogens are unable to grow at pH 3.5. Data represent 3 biological replicates (the average of 3 technical replicates). Error represents mean +/− SD. (D) Biofilm opacity measured by optical density (OD) at 600 nm after 48 h of static growth seeded at physiological pH (*N* = 6, each the sum of 3 technical replicates). Statistical analysis represents a Friedman test with Dunn's multiple comparisons test. (E) Metabolic activity within the biofilm after 48 h of static growth seeded at physiological pH, measured by the production of formazan from MTT at OD 570 nm. Error represents mean +/− SD. (F) Growth of *C. rodentium* at physiological pH in DMEM base media over 20 h. Data represent 3 biological replicates (the average of 3 technical replicates). Error represents mean +/− SD. (G) Attachment of *C. rodentium* to CMT-93 murine colonic epithelial cells after preinduction in DMEM base media adjusted to gastrointestinal pH. Data represent biological replicates (the average of 3 technical replicates). (H) Attachment of wild-type (WT) or T3SS-deficient (Δ*escN*) *C. rodentium* to colonic epithelial cells after preinduction at gastrointestinal pH. Data represent biological replicates (the average of 3 technical replicates). (I) Representative images of actin pedestal formation on CMT-93 cells following infection with *C. rodentium* pre-induced at pH 4.8 and pH 7.0 (*N* = 3 biological replicates). Scale bar represents 30 μm. Summary data displayed in S2 Fig can be found in S1 Data.
(TIF)

**S3 Fig. Homology to the *E. coli* CadA lysine decarboxylase.** (A) Protein sequence homology between *C. rodentium* LdcC and *E. coli* CadA. (B) Survival of wild-type and Δ*ldcC C. rodentium* strains in gastric pH 3.5 over the range of average gastric emptying time. Each point represents the average of 3 biological replicates (3 technical replicates per biological replicate). Statistical analysis represents a two-way ANOVA with Geisser–Greenhouse correction. Summary data displayed in S3 Fig can be found in S1 Data.
(TIF)

**S4 Fig. *C. rodentium* survival at gastric pH.** (A) Survival of *C. rodentium* pre-induced aerobically at either neutral pH 7 or colonic pH 5.7 for 3.5 h (to mid-log phase) before exposure to gastric pH 3.5. Each point represents the average of 3 biological replicates (3 technical replicates per biological replicate). Statistics represent a mixed-effects model with Geisser–Greenhouse correction and Šidák's multiple comparisons test. Error represents mean +/− SD. (B) Survival of *C. rodentium* shed in the feces on day 2 postinfection (pi) at gastric pH 3.5. Fecal pellets were not disrupted before submersion in LB pH 3.5. Statistical analysis represents a Mann–Whitney test (*N* = 6–7). Summary data displayed in S4 Fig can be found in S1 Data.
(TIF)

**S5 Fig. Factors involved in gastric acid secretion postinfection.** (A) Spearman correlation of stomach pH and weight of stomach contents (*N* = 23). Line represents linear regression +/− 95% confidence interval. *p*- and rho-values represent Spearman correlation values. (B)

Representative stomach images from fasted control and day 8-infected mice. Scale bar represents 1 cm. (C) Normalized fold expression of genes associated with host regulation of acid secretion within stomach tissue from naïve and infected mice ($N$ = 4–8). Statistics represent a two-way ANOVA with Šidák's multiple comparisons test. (D) Normalized fold expression of genes associated with host negative regulation of acid secretion within stomach tissue from naïve and infected mice ($N$ = 7–8). Statistics represent a two-way ANOVA with Šidák's multiple comparisons test. Summary data displayed in S5 Fig can be found in S1 Data.
(TIF)

**S6 Fig. The relationship of stomach pH with infection, epithelial barrier integrity, and inflammation.** (A) Spearman correlation of stomach pH and fecal burden of *Salmonella* Typhimurium at day 4 of a murine typhoid infection model ($N$ = 14). Line represents linear regression +/− 95% confidence interval. *p*- and rho-values represent Spearman correlation values. (B) pH of drinking water used in the DSS-colitis model. (C) Colonic expression of *TJP1* in naïve and *C. rodentium*-infected mice at day 8 postinfection ($N$ = 4–8). Statistics represent a Mann–Whitney test. (D) Serum calprotectin levels measured by ELISA in naïve and *C. rodentium*-infected mice at day 8 postinfection ($N$ = 4–8). Statistics represent a Mann–Whitney test. (E) Spearman correlation of stomach pH and serum calprotectin levels of naïve and infected mice ($N$ = 20). Line represents linear regression +/− 95% confidence interval. *p*- and rho-values represent Spearman correlation values. (F) Cytokine levels measured in the colons of naïve mice or mice colonized with Δ*escN C. rodentium*, commensal *E. coli* Mt1B1, or wild-type (WT) *C. rodentium* at day 4 postinoculation, or WT *C. rodentium* at day 8 postinfection, or mice treated with dextran sodium sulfate (DSS) to induce colitis ($N$ = 4–8). (G) Spearman correlation of colonization levels by avirulent Δ*escN C. rodentium* and commensal isolate *E. coli* Mt1B1 with stomach pH ($N$ = 10). Line represents linear regression +/− 95% confidence interval. *p*- and rho-values represent Spearman correlation values. (H) Cecum and colon bacterial burden at day 4 post-colonization with WT *C. rodentium*, Δ*escN C. rodentium*, or commensal *E. coli* Mt1B1. Statistics represent a two-way ANOVA with Šidák's multiple comparisons test. Summary data displayed in S6 Fig can be found in S1 Data.
(TIF)

**S1 Data. Raw data used for the generation of graphs presented in Figs 1–7 and S1–S6.**
(XLSX)

**S1 Uncropped Images. Raw uncropped images representing the analysis shown in Figs 2, 4, S2, and S5.**
(PDF)

**S1 Table. Bacterial strains used in this study.**
(DOCX)

**S2 Table. Oligonucleotide primers and plasmids used for making deletion constructs of *C. rodentium ldcC.***
(DOCX)

**S3 Table. Oligonucleotide primers used for RT-qPCR.**
(DOCX)

## Acknowledgments

The authors thank all of our colleagues in the Finlay laboratory for their support and assistance, especially L Thorson and T Bozorgmehr. Supporting images were created with

BioRender.com under agreements HD270MIETZ ([Fig 1A]) and RE270RB9H8 ([Fig 7C]). Imaging was performed in the LSI Imaging Core Facility (RRID:SCR_023783) at the University of British Columbia.

## Author Contributions

**Conceptualization:** Sarah E. Woodward.

**Data curation:** Sarah E. Woodward, Laurel M. P. Neufeld, Jorge Peña-Díaz.

**Formal analysis:** Sarah E. Woodward, Laurel M. P. Neufeld, Jorge Peña-Díaz, Wenny Feng.

**Funding acquisition:** B. Brett Finlay.

**Investigation:** Sarah E. Woodward, Laurel M. P. Neufeld, Jorge Peña-Díaz, Wenny Feng, Antonio Serapio-Palacios, Isabel Tarrant, Wanyin Deng.

**Methodology:** Sarah E. Woodward, Laurel M. P. Neufeld, Jorge Peña-Díaz, Antonio Serapio-Palacios.

**Supervision:** B. Brett Finlay.

**Validation:** Sarah E. Woodward, Laurel M. P. Neufeld, Jorge Peña-Díaz, Wenny Feng.

**Visualization:** Sarah E. Woodward, Jorge Peña-Díaz, Wenny Feng, Antonio Serapio-Palacios.

**Writing – original draft:** Sarah E. Woodward.

**Writing – review & editing:** Sarah E. Woodward, Laurel M. P. Neufeld, Jorge Peña-Díaz, Wenny Feng, Antonio Serapio-Palacios, Isabel Tarrant, Wanyin Deng, B. Brett Finlay.

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
