## [Editor Report · Decision Letter 0]

15 Jan 2024

Dear Dr Finlay, 

Thank you for submitting your manuscript entitled "Pathogen and host adapt pH responses during enteric infection" for consideration as a Research Article by PLOS Biology.

Your manuscript has now been evaluated by the PLOS Biology editorial staff, as well as by an academic editor with relevant expertise, and I am writing to let you know that we would like to send your submission out for external peer review.

Once your full submission is complete, your paper will undergo a series of checks in preparation for peer review. After your manuscript has passed the checks it will be sent out for review. To provide the metadata for your submission, please Login to Editorial Manager (https://www.editorialmanager.com/pbiology) within two working days, i.e. by Jan 17 2024 11:59PM.

Kind regards,

Melissa

Melissa Vazquez Hernandez, Ph.D.

Associate Editor

PLOS Biology

---

## [Decision Letter · Decision Letter 1]

13 Mar 2024

Dear Dr Finlay,

I would like to apologize for the duration of the revision process. Thank you for your patience while your manuscript "Pathogen and host adapt pH responses during enteric infection" was peer-reviewed at PLOS Biology. It has now been evaluated by the PLOS Biology editors, an Academic Editor with relevant expertise, and by several independent reviewers. 

In light of the reviews, which you will find at the end of this email, we would like to invite you to revise the work to thoroughly address the reviewers' reports. As you will see below, specifically Reviewer #2 and 3 have concerns regarding the lack of support of some conclusions. Reviewer #2 suggests additional experiments such as testing infection outcomes in LB and following a transcriptomics approach, as well as additional in vivo experiments. Reviewer #3 asks to demonstrate the reduced acid tolerance using an LdcC mutant and to quantify bacterial burden across groups.

IMPORTANT: although Reviewer #2’s suggestions regarding additional transcriptomic analysis and in vivo experiments could potentially strengthen your study, after discussion with the Academic Editor and the reviewers, we determined to leave to your discretion to follow such suggestions. Additionally, the Academic Editor has provided additional requests regarding presentation and clarification of some points in the text and choice of experiments (see the foot of this email). Therefore, with the exception of additional in vivo experiments, addressing the concerns raised by all reviewers and the Academic Editor is necessary for your manuscript to be considered for publication in PLOS Biology.

Given the extent of revision needed, we cannot make a decision about publication until we have seen the revised manuscript and your response to the reviewers' comments. Your revised manuscript is likely to be sent for further evaluation by all or a subset of the reviewers.

**IMPORTANT - SUBMITTING YOUR REVISION**

*Re-submission Checklist*

*Published Peer Review*

*PLOS Data Policy*

*Blot and Gel Data Policy*

Sincerely,

Melissa

Melissa Vazquez Hernandez, Ph.D.

Associate Editor

PLOS Biology

REVIEWERS' COMMENTS

Reviewer #1: 

In this manuscript, Woodward et al. examined the effects of pH on the pathogen Citrobacter rodentium. The authors found that pH mirroring different regions of the gastrointestinal tract differentially regulated virulence gene expression and cell morphology. The authors supported their findings with well-designed animal models. This work is very timely as the intestinal environment, including the pH, is not often assessed in animal models of infection. The manuscript is well-written, the figures are excellent and overall, the work is very compelling. The authors did an excellent job addressing the concerns of the original reviewers. As a new reviewer, I only have two minor comments. 

Comments:

1. I am surprised to see a pH of 5.4 in the ileum. This pH is a bit lower compared to other stains of mice- particularly FvBN mice. I think Figure 1A needs to be modified to include more of the literature that measures intestinal pH along the length of the murine gastrointestinal tract in other animals strains. The current literature appears to only covers the pH of humans. Here are some pertinent references: PMID: 9662405, PMID: 24072680, PMID: 33670214, PMID: 18088506, PMID: 24429819, PMID: 10480906. The majority of these studies observed a pH~7 in the ileum. However, the data looks solid in the manuscript, and I trust that the authors are correct in reporting a pH of ~5 in the small intestine of the BALB/c mice.

2. I think it's interesting that C. rodentium elevates the pH of the intestine, when the same effect is observed in humans in response to C. difficile infection (PMID: 25552580). This point could be added to the discussion to strengthen the paragraph on pH in humans. 

Reviewer #2: 

This paper has been reviewed before, so I am not going to repeat the general summary. It is well written and the discussion is informative. 

The original reviewers raised some concerns and the authors, in most parts, provided satisfactory responses. Nonetheless, the paper remains somewhat descriptive. 

The authors first determined the regional pH along the gut. They then conducted phenotypic and gene expression characterizations in the different pHs. However, the conclusions were not tested in vivo. In the last two figures the authors focused on the impact of colitis on the gastric pH. For me, Fig. 1-5 and Fig. 6-7 are disjointed, the paper seems to report two stories in one. 

Specific comments:

1. The explanation of the observed increase in biomass at pH 3.5 in the absence of metabolic activity is not clear ("encapsulation of community members within biofilms").

2. While the authors tested cell attachment, they haven't quantified actin polymerization in CMT-93 cells. In their rebuttal they justified this by citing Liang et al, PNAS 2023. However, this paper reports actin straining in CMT-93 cells. 

3. To translate the data to infection, it might be useful to test infection outcomes of C. rodentium grown in LB in different pH's. Moreover, a kfc mutant should be tested in vivo. 

4. In Figs. 3 and 4 the authors report expression of multiple genes across the pH range. While selecting few rationale targets, a transcriptomic approach would provide an unbiased readout. Moreover, the authors could probably use these data as a guide to generate specific mutants for in vivo testing. Otherwise, the relevance of the measurements remains elusive. 

5. The rationale for measuring changes to gastric pH and expression of host gene in response to C. rodentium infection is not apparent, considering that at days 4 and 8 pi C. rodentium is found further down the GI tract. Nonetheless, as the change in gastric pH seems to be a generic response to tissue damage, this is a novel discovery. However, the gastric data seem an add-on to the C. rodentium responses to a pH gradient described in earlier parts of the paper. The authors should better integrate these two "stories". 

6. I thought the authors should have quantified shedding of the wild type, mutant and commensal E. coli strains in the last experiment.

Reviewer #3: 

In this work Woodward et al study the effect of difference pH levels on the murine bacterial pathogen Citrobacter rodentium, a model of human attaching and effacing pathogens. They assessed the pH levels throughout the gastrointestinal tract and conducted various in vitro assays to evaluate bacterial responses under conditions that simulate the intestinal environment. They also determined host acid-related characteristics and responses. The topic is interesting and some of the observations, such as pH measurements across the GI tract, and host gastric pH under infection and inflammation are important and interesting and will definitely contribute the field. However, some of the major claims are overreaching and lack sufficient data support.

Comments:

Lines 232-241 - The major claim of growth arrest at pH 3.5 is well supported, but it is unclear to me how the authors distinguish growth and biofilm formation. Both assays appear to be reading absorbance at 600nm. Also, metabolic activity and biofilm production were measured after 48 hours of incubation, at which the bacteria are probably at deep stationary phase, although the transit time in the small intestine is a few hours. Cell morphology dramatically change much faster, as the authors show in Fig4. Other readouts were also taken much earlier.

Epithelial attachment - Actin rearrangements and pedestal formation are the hallmark of virulence for A&E pathogens. Although the escV mutant demonstrates reduced adherence to CMT-93 cells, highlighting the dependence on TTSS, it remains unclear why the authors did not utilize a cell line that would allow direct observation and quantification of pedestal formation.

Bacterial stress responses - The claim that C. rodentium uses amino acid decarboxylation to combat acid induced stress is an overstatement, given the limited supporting data. Demonstrating reduced acid tolerance in a mutant LdcC strain would provide more convincing evidence for this claim.

The increase in atpD expression in pH4.4 - it seems to mostly rely on 1 data point. An outlier test should be conducted here, and in Figure 4C (for degP).

Line 337 - unclear. What is "fluid regulation of membrane protein synthesis"?

Fig 6B - I'm not a statistician, but at a glance, there appears to be a low correlation between the data points and the shaded area. Same correlation in Fig. 7B looks convincing. 

Lines 452-454 - Bacterial burden should be quantified across the experimental groups by plating, qPCR or any other valid method.

COMMENTS FROM THE ACADEMIC EDITOR:

1. The authors could consider bringing FigS1 C to the main Fig 1 as a new panel. Even if they don’t take this suggestion, the legend of this figure panel should be clarified regarding how many BL6 mice and how many BALBc mice. Currently is just says N=8-18 but does not state the sample size for each genetic background.

2. In the section host-adaptation, page 16 line 352, the authors should clarify if they mean that C. rodentium adapts evolutionary or if to physiological adaptation.

3. In the section Gastric PH decreases in response to further infection, the authors should state why they chose to measure pH at day 4 and at day 8 as done in Fig 6A. I am not asking for more experiments just for a rational of the choice.

---

## [Editor Report · Decision Letter 2]

16 Jul 2024

Dear Dr Finlay,

Thank you for your patience while we considered your revised manuscript "Pathogen and host adapt pH responses during enteric infection" for publication as a Research Article at PLOS Biology. This revised version of your manuscript has been evaluated by the PLOS Biology editors and the Academic Editor.

Based on our Academic Editor's assessment of your revision, we are likely to accept this manuscript for publication, provided you satisfactorily address the remaining editorial points. Please also make sure to address the following data and other policy-related requests.

a) We routinely suggest changes to titles to ensure maximum accessibility for a broad, non-specialist readership, and to ensure they reflect the contents of the paper. In this case, we would suggest a minor edit to the title, as follows. Please ensure you change both the manuscript file and the online submission system, as they need to match for final acceptance:

"Both pathogen and host adapt dynamically to pH changes along the intestinal tract during enteric bacterial infection"

b) Given that some of the explanations given in the point by point response would be good for all readers to have access to, we would encourage that you opt into the transparent peer-review option, where editor letters and reviewer reports are published.

c) We do not allow Supplementary Materials and Methods, as well as Supplementary References. Please place your current Supplementary Materials and Methods, and their references, in the main methods section and main reference sections, respectively. The Supplementary Tables can remain there but they should be called out approapiately in the newly augmented "Materials and Methods" section.

d) Thank you for supplying the numerical values of most of the figures. However, we are missing the values for Figures SB3 and S6H. We have also noted that the data for Figure S4B seems to be mislabelled.

e) Please note that per journal policy, the model system/species (mouse) studied should be clearly stated in the abstract of your manuscript. Thank you for already providing the name of Citrobacter rodentium.

f) Please ensure that your Data Statement in the submission system accurately describes where your data can be found and is in final format, as it will be published as written there.

g) Per journal policy, if you have generated any custom code during the course of this investigation, please make it available without restrictions upon publication. Please ensure that the code is sufficiently well documented and reusable, and that your Data Statement in the Editorial Manager submission system accurately describes where your code can be found.

We expect to receive your revised manuscript within two weeks. 

*Published Peer Review History*

*Press*

Sincerely,

Melissa

Melissa Vazquez Hernandez, Ph.D.

Associate Editor

PLOS Biology

---

## [Editor Report · Decision Letter 3]

19 Jul 2024

Dear Dr Finlay and Sarah,

Thank you for the submission of your revised Research Article "Both pathogen and host dynamically adapt pH responses along the intestinal tract during enteric bacterial infection" for publication in PLOS Biology. On behalf of my colleagues and the Academic Editor, Isabel Gordo, I am pleased to say that we can in principle accept your manuscript for publication, provided you address any remaining formatting and reporting issues. These will be detailed in an email you should receive within 2-3 business days from our colleagues in the journal operations team; no action is required from you until then. Please note that we will not be able to formally accept your manuscript and schedule it for publication until you have completed any requested changes.

IMPORTANT: regarding the transparent peer review, this will be given as an option in the following process. 

PRESS

Sincerely, 

Melissa

Melissa Vazquez Hernandez, Ph.D., Ph.D.

Associate Editor

PLOS Biology
